# Heterogeneous trajectories of exercise self-efficacy and its predictors in patients with multivessel coronary artery disease: A longitudinal study

Binbin Sun[1], Jin Wang[2], Haijiao Xiao[3], Yutong Wang[3], Jianhui Wang[ID][2]*

**1** Tianjin Medical University Cancer Institute & Hospital, National Clinical Research Center for Cancer, Tianjin's Clinical Research Center for Cancer, Key Laboratory of Cancer Prevention and Therapy, Tianjin, China, **2** Nurse Administration Department, Tangshan Gongren Hospital, Tangshan, Hebei Province, China, **3** Nursing and Rehabilitation College, North China University of Science and Technology, Tangshan, Hebei Province, China

* anita30@163.com

## Abstract

### Objective

The developmental trajectory of exercise self-efficacy refers to the course of change in an individual's belief in their capability to successfully perform exercise-related behaviors over time. This study aims to explore the developmental trajectory of exercise self-efficacy in patients with multivessel coronary artery disease, and analyze the predictors of various trajectory subgroups.

### Methods

Between September 2023 to October 2024, 297 patients with multivessel coronary artery disease were recruited from three tertiary hospitals in Tangshan, China. Exercise self-efficacy was measured using the Multidimensional Self-Efficacy for Exercise Scale on the third day of admission (T1), one month after discharge (T2), three months after discharge (T3), and six months after discharge (T4). The latent class growth model was employed to identify the developmental trajectory of exercise self-efficacy in patients with multivessel coronary artery disease. Multinomial logistic regression was adopted to determine the predictors of trajectory subgroups.

### Results

Three distinct trajectories of exercise self-efficacy were identified: the "Low-Efficacy Decline Group" (22%), the "High-Efficacy Ascending-Stable Group" (34%), and the "Moderate-Efficacy Continuous Increase Group" (44%). Multinomial logistic regression analysis revealed significant determinants of distinct exercise self-efficacy trajectories, when compared with the moderate-efficacy continuous increase group,

**Data availability statement:** All relevant data are within the paper and its Supporting Information files.

**Funding:** This study was supported by the Medical Science Research Project of Hebei in the form of a grant awarded to J.W. (Grant No.20241512) and the Tianjin Key Medical Discipline Construction Project in the form of a salary for B.S. (Grant No.TJYXZDXK-3-003A). The specific roles of this author are articulated in the 'author contributions' section. The funders had no role in study design, data collection and analysis, decision to publish, or preparation of the manuscript.

**Competing interests:** The authors have declared that no competing interests exist.

the predictors for the low-efficacy decline group included having diabetes, a lack of exercise habits, low social support, and anxiety. In contrast, the predictors for the high-efficacy ascending-stable group were high average monthly household income, established exercise habits, and strong social support.

## Conclusion

The study revealed heterogeneous trajectories of exercise self-efficacy among patients with multivessel coronary artery disease, highlighting the necessity for personalized intervention strategies. These findings offer a valuable opportunity for early prevention and targeted interventions aimed at enhancing exercise self-efficacy.

## 1. Introduction

Coronary artery disease is one of the leading causes of death and disability globally [1]. With advancements in diagnostic and treatment technologies, patients with multivessel coronary artery disease are increasingly identified through coronary angiography, with an incidence ranging from 40% to 70% [2]. Multivessel coronary artery disease refers to coronary atherosclerosis that affects at least two major coronary vessels, each exhibiting stenosis of 50% or greater, which leads to a significant reduction in blood flow to the heart [3]. This condition is associated with severe angina pectoris, malignant arrhythmias, and even the risk of sudden cardiac death [4]. Multivessel coronary artery disease not only causes significant and lasting damage to patients' heart function but also greatly diminishes their quality of life. Despite evidence that exercise rehabilitation improves cardiovascular function, exercise capacity, and quality of life [5,6], existing studies indicate that the current state of exercise rehabilitation across various countries is concerning. According to statistics, the average participation rate of exercise rehabilitation in Europe is 30%, while the participation rate in the United States ranges from 20% to 30% [7]. Wu et al. discovered that the participation rate in China was only 13.04% [8], far below the minimum level recommended by cardiac rehabilitation guidelines for effective cardiac recovery. The low participation rate and poor compliance with exercise rehabilitation in patients have become critical clinical challenges, profoundly impacting their rehabilitation outcomes and long-term prognosis. Therefore, it is essential to explore effective strategies for enhancing patients' active participation in exercise rehabilitation.

Exercise self-efficacy refers to an individual's subjective assessment of their ability to successfully engage in physical activities. Research has confirmed that it enhances patients' confidence in exercising and improves their adherence to exercise rehabilitation [9]. Exercise self-efficacy reflects the level of confidence individuals possess in the context of physical activity, influencing their determination in setting exercise goals, their persistence in maintaining an exercise routine, and the effort they are willing to invest [10]. However, for patients with multivessel coronary artery disease, whose exercise behaviors are often constrained by fear of disease

recurrence, physical debilitation, and psychological stress, understanding the dynamic changes in exercise self-efficacy over time is essential to formulate targeted intervention strategies.

Self-efficacy theory highlights that exercise self-efficacy is a psychological construct that exhibits dynamic characteristics [10]. Current studies on exercise self-efficacy are primarily limited to cross-sectional surveys, which typically capture data at a specific time point to assess the overall level of exercise self-efficacy [11,12]. Alternatively, some studies employ repeated measures to investigate the dynamics of exercise self-efficacy [13], while often overlooking group heterogeneity. It should be noted that not all individuals exhibit the same developmental trajectory of exercise self-efficacy. Previous research has examined self-efficacy trajectories in different populations. Young et al. identified four distinct self-efficacy trajectories in patients with multiple sclerosis, with declining, slightly declining, stable or improving self-efficacy [14]. Liu et al. conducted a trajectory study on maternal pelvic floor muscle training self-efficacy and identified four potential trajectory subgroups: high functioning group, developmental group, blocked group, and low functioning group [15]. These results indicate that individual differences should be considered in longitudinal studies. In addition, numerous studies have investigated the factors associated with exercise self-efficacy but lack systematization. Patient demographic and clinical characteristics significantly correlate with exercise self-efficacy, including gender, age, education level, occupation, marital status, exercise habits, and comorbid conditions [11,16–19]. However, few studies have examined the changes in exercise self-efficacy among patients with multivessel coronary artery disease at various time points, and there has been limited attention given to the influence of related factors on the exercise self-efficacy trajectory.

In conclusion, the development trajectory of exercise self-efficacy in patients with multivessel coronary artery disease has not been thoroughly investigated. The influencing factors for the various trajectory subgroups remain unclear. In light of these knowledge gaps, the present study aimed to explore the developmental trajectory of exercise self-efficacy in patients with multivessel coronary artery disease using a latent class growth model, and identify baseline factors influencing these trajectories.

## 2. Methods

### 2.1. Study design and participants

This longitudinal study was reported by Strengthening the Reporting of Observational Studies in Epidemiology (STROBE, S1 File) and registered in the Chinese Clinical Registry Center (No. ChiCTR2300076995).

This study was conducted from 25 September 2023–01 October 2024 in three tertiary hospitals in Tangshan, China. Inclusion criteria were as follows: age 18 years or older; met the diagnostic criteria for coronary artery disease; ≥2 major coronary arteries with ≥ 50% luminal stenosis confirmed by coronary angiography. Exclusion criteria included severe dysrhythmias (such as ventricular fibrillation and third-degree atrioventricular block), and hearing or communication disabilities that hindered the successful completion of questionnaires.

### 2.2. Ethical statement

The study adhered to the principles outlined in the Declaration of Helsinki. Written informed consent was obtained from all patients before data collection, and the study was approved and registered with the Tangshan Gongren Hospital Ethics Committee (GRYY-2023–116).

### 2.3. Sample size

The sample size determination was conducted using G*power 3.1 software. Given that our study involved tracking exercise self-efficacy trajectories derived from four repeated measurements, we employed the Single-group repeated-measures analysis of variance algorithm [20]. The nonsphericity correction (ε) was set to 0.5, a value widely utilized in repeated-measures designs when the precise correlation structure between repeated measurements is not fully

established in advance. This value accounts for a moderate deviation from sphericity, thus ensuring the analysis remains robust across diverse potential correlation patterns among the four measurement points. The statistical power was set to 0.8, which strikes a balance between the risk of Type II errors and the practicality of recruiting participants, guaranteeing the study a reasonable likelihood of identifying meaningful trajectory patterns if they exist. With 95% confidence intervals (CI), these parameters resulted in a minimum required sample size of 182. Considering a potential 20% attrition rate, a total of 219 participants were deemed necessary to ensure sufficient statistical power for analyzing the exercise self-efficacy trajectories across the four measurement points.

## 2.4. Measures

**2.4.1. Demographic and clinical characteristics.** Electronic medical records were utilized to gather patient demographic characteristics and clinical data. The demographic characteristics included gender, age, education level, marital status, residential type, employment status, residential location, average monthly household income, and exercise habits. Clinical data included the treatment method, family history of coronary artery disease, hypertension, diabetes, hyperlipidemia, and stroke. In this study, we defined the participants as having an exercise habit if they engaged in physical activity for a cumulative duration of at least 30 minutes per day on at least three days per week within the three months prior to hospitalization, including aerobic exercises such as walking, jogging, swimming, and cycling, as well as anaerobic exercises mainly consisting of low-load resistance training.

**2.4.2. Exercise self-efficacy.** Exercise self-efficacy was assessed using the Multidimensional Self-Efficacy for Exercise Scale (MSES). Developed by Rodgers in 2008 [21], the scale was later translated into Chinese [11]. It includes three dimensions: task efficacy, coping efficacy, and scheduling efficacy, with a total of nine items. Task efficacy refers to an individual's confidence in mastering the elemental aspects of exercise (e.g., using proper exercise techniques, following instructor directions, and performing all required movements); coping efficacy denotes an individual's confidence in sustaining exercise participation under challenging circumstances (e.g., exercising when feeling discomfort, lacking energy, or unwell); and scheduling efficacy represents an individual's confidence in integrating exercise into daily life despite time constraints (e.g., including exercise in daily routines, consistently exercising at least three times a week, and arranging schedules to ensure regular exercise). Each item is scored from 0 (no confidence) to 10 (full confidence), and yielding total scores ranging from 0 to 90. Higher scores indicate greater confidence in engaging in exercise.

**2.4.3. Social support.** Social support was assessed using the Exercise Social Support Scale (ESSS), which was developed by Salis et al. [22] and later simplified by Hankonen et al. [23]. The scale consists of five items, with responses ranging from 1 (never) to 5 (always) for each day, resulting in a total score that ranges from 5 to 25. Higher scores indicate greater social support for exercise.

**2.4.4. Anxiety.** The Generalized Anxiety Disorder Questionnaire-7 (GAD-7) is a seven-item self-report instrument designed to assess the severity of generalized anxiety disorder and related anxiety symptoms [24]. Each item prompts the individual to rate the severity of their symptoms over the past two weeks using a four-point Likert scale, with possible responses ranging from 0 (Not at all) to 3 (Nearly every day). Total scores can range from 0 to 21, with higher scores indicating a greater level of anxiety symptoms.

## 2.5. Data collection

The research team obtained permission from department heads at three hospitals before recruiting participants based on established inclusion and exclusion criteria. Trained and qualified researchers conducted the investigation. Specifically, the researchers were master's degree candidates in nursing, who met the required qualifications by completing special-ized training in clinical research methods. This training covered ethical principles in human subject research, standard-ized data collection procedures, and effective patient communication—all essential for conducting clinical investigations involving patient populations. Before the survey, the purpose, significance, and confidentiality measures were explained

to the participants, and the survey proceeded only after obtaining their informed consent. Data were collected on the third day of admission (T1), one month after discharge (T2), three months after discharge (T3), and six months after discharge (T4). Survey timing was guided by consensus [25,26], the transtheoretical model [27], and prior longitudinal study [28]. Phase I cardiac rehabilitation typically begins within 24 hours of admission but may be delayed by three to seven days in patients with unstable conditions (e.g., hemodynamic instability, severe arrhythmias) [25]. We chose the third day after admission (T1) for the initial survey based on patients' conditions and clinical feasibility. Phase II rehabilitation initiates one month after discharge [25], and consensus emphasizes exercise capacity assessments at three months and six months after discharge [26]. The transtheoretical model, a significant theory for understanding individual behavior change, posits that behavior change is a gradual process and identifies one month and six months as critical time points for individual behavior change [27]. A previous study, which focused on behavior patterns and physical function in stroke patients, found that the time points for longitudinal investigations of patients were predominantly established during hospitalization, as well as at one month, three months, and six months after discharge [28]. Consequently, we selected follow-up periods of one month (T2), three months (T3), and six months (T4) after discharge to explore the short- to medium-term changes in exercise self-efficacy in patients with multivessel coronary artery disease. Baseline data (T1) was collected through face-to-face surveys. Patients completed the questionnaire themselves. If patients found it inconvenient to fill out the questionnaire, the investigator would read the questions and answer options recording the patient's responses accordingly. Completed questionnaires were immediately checked for completeness and logical consistency. With patient and family consent, at least two commonly used telephone numbers were retained. For subsequent follow-up assessments at T2 to T4, trained researchers administered the questionnaires via telephone at the predefined time points after discharge.

## 2.6. Data analysis

Mplus 8.3 software was used for latent class growth analysis, with model fitting evaluated using the following criteria: Akaike Information Criterion (AIC), Bayesian Information Criterion (BIC), and Sample-Size Adjusted BIC (aBIC), where lower values indicate better fit; the entropy value was used to assess classification accuracy (values closer to 1 indicate more precise trajectory differentiation); the Bootstrapped Likelihood Ratio Test (BLRT) and Vuong-Lo-Mendell-Rubin Likelihood Ratio Test (VLMRT) were used to determine the optimal number of classes, with significant results for both tests indicating that the k-class model was superior to the (k-1)-class model [29]. SPSS 22.0 software was used for data analysis. Measurement data conforming to the normal distribution were described by mean ± standard deviation. For univariate analysis, the chi square test was used for categorical variables, and the ANOVA test was used for continuous variables. Multinomial logistic regression model was used to examine the effects of variables on the latent trajectory subgroups. Statistical significance was set at $P < 0.05$.

## 3. Results

### 3.1. Characteristics of the participants

A total of 371 patients were enrolled in the study, with 74 excluded for the following reasons: death ($n = 1$), worsening illness ($n = 14$), poor cooperation ($n = 26$), and loss of contact due to phone rejection or busy signals ($n = 33$). Ultimately, 297 patients were included in the analysis. A STROBE-compliant flow diagram illustrating participant recruitment and follow-up is presented in Fig 1. To address potential bias from attrition, we compared baseline characteristics between included and excluded patients with multivessel coronary artery disease, and no statistically significant differences were observed (all $P > 0.05$; S2 Table). To further verify the robustness of these findings against missing data, we performed a sensitivity analysis. Little's MCAR test confirmed that the missing follow-up data conformed to the missing at random assumption ($P = 0.312$), indicating that attrition did not introduce substantial bias into the results. Among the participants, there were 99 females (33.3%) and 198 males (66.7%). The mean age of the patients was 62.08 ± 10.27 years. In terms of education,

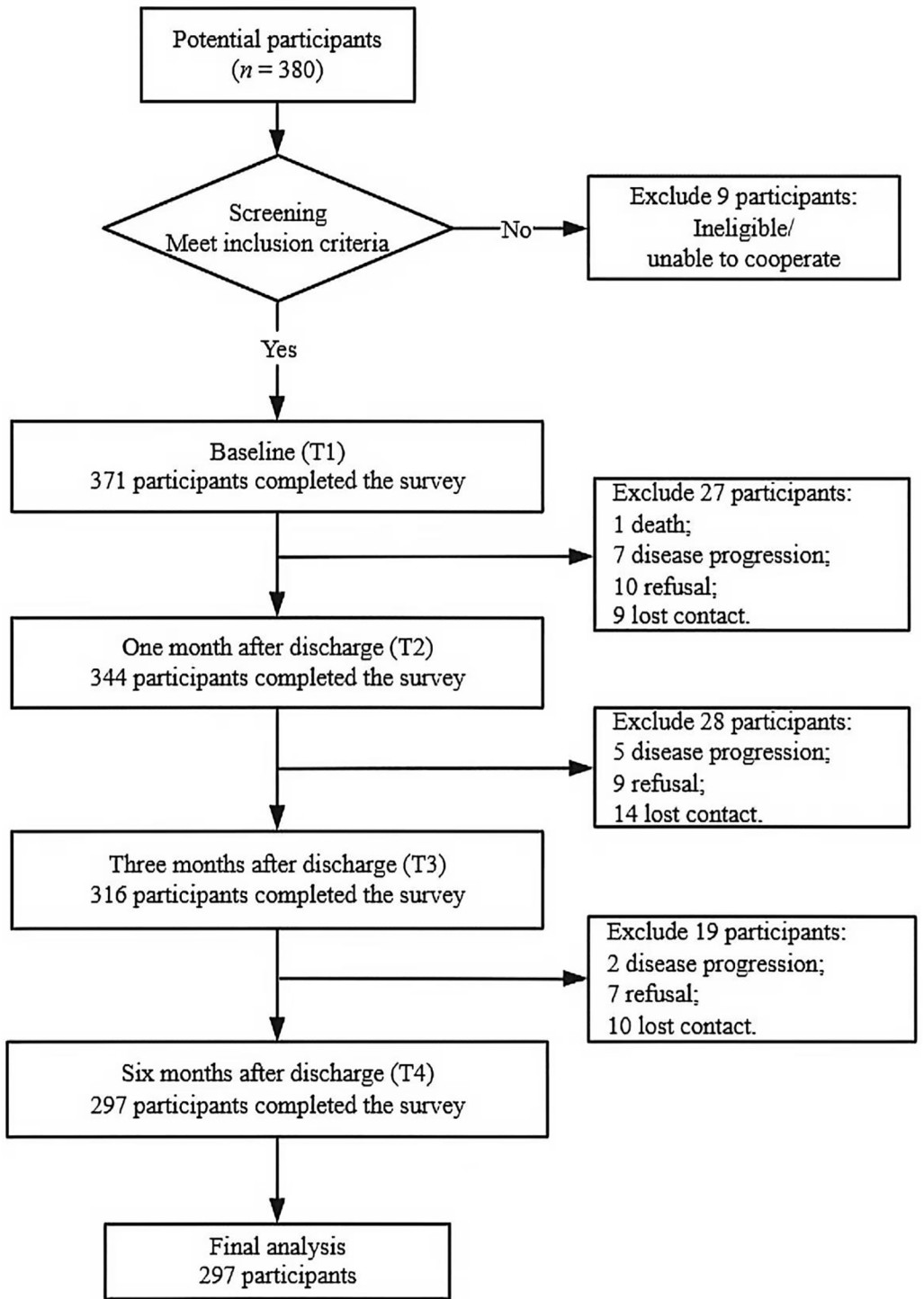

**Fig 1. STROBE flow diagram of participant recruitment and follow-up.** The participant recruitment and follow-up process involved 380 potential participants: 9 excluded at screening, 371 enrolled at baseline (T1), and 297 remained for final analysis after follow-up exclusions (27 at T2, 28 at T3, 19 at T4).

26.9% had completed elementary education (≤6 years), 47.8% had secondary education (7–9 years), and 25.3% had higher education (>9 years). Most patients (94.6%) were married, and 51.9% were employed, with 57.6% residing in urban areas. Additionally, 45.5% reported having an exercise habit. Furthermore, 48.5% had a family history of coronary artery disease, 62.6% had hypertension, 33.3% had diabetes, 6.1% had hyperlipidemia, and 7.7% had stroke.

### 3.2. Identifying exercise self-efficacy trajectories

The latent class growth model fitting analysis was conducted by sequentially extracting between one and five classes. The results showed that the model fit indices—AIC, BIC, and aBIC—decreased progressively as the number of classes increased, indicating a consistent improvement in model fit. The three-class model had the highest entropy value (0.922), and both the VLMRT and BLRT were statistically significant ($P < 0.001$). To validate the reliability of the three-class trajectory model, a 10-fold cross-validation was performed as the sensitivity analysis. The full dataset was randomly split into a training set (70% of samples) and a validation set (30% of samples) for 10 independent iterations (S3 Table). These findings indicate that the three-class model provided the best fitting results and was more accurate than the other models. Thus, after comprehensive evaluation of model fit and clinical relevance, the three-class model was selected as the final specification for exercise self-efficacy trajectories in patients with multivessel coronary artery disease. Model fit indices are presented in Table 1.

Based on intercept/slope estimates and Bonferroni-corrected multiple comparisons, three distinct trajectories were identified (Tables 2 and 3): Class 1 had low initial self-efficacy (intercept = 25.217, $P < 0.001$) with a gradual decline over time (slope = −0.965, $P < 0.001$), showing no significant changes between T1–T2 or T1–T3 but significant declines at other time points. Therefore, Class 1 was designated as the "Low-Efficacy Decline Group"; Class 2 exhibited higher initial self-efficacy (intercept = 44.561, $P < 0.001$) with an upward trend (slope = 6.076, $P < 0.001$), showing no significant changes

**Table 1. Model fit indices for one to five exercise self-efficacy trajectories.**

| Model | AIC | BIC | aBIC | Entropy | VLMRT (P value) | BLRT (P value) | Latent trajectory proportions (%) |
|---|---|---|---|---|---|---|---|
| 1 | 8764.575 | 8801.512 | 8769.798 | 1 | – | – | – |
| 2 | 8465.157 | 8516.869 | 8472.471 | 0.901 | <0.001 | <0.001 | 0.26/0.74 |
| 3 | 8185.261 | 8251.248 | 8194.664 | 0.922 | <0.001 | <0.001 | 0.22/0.34/0.44 |
| 4 | 8086.276 | 8164.538 | 8097.768 | 0.911 | 0.006 | <0.001 | 0.21/0.26/0.17/0.36 |
| 5 | 8030.073 | 8126.110 | 8043.655 | 0.903 | 0.258 | <0.001 | 0.16/0.19/0.29/0.24/0.12 |

Abbreviation: aBIC, Sample-Size Adjusted BIC; AIC, Akaike Information Criterion; BIC, Bayesian Information Criterion; BLRT, Bootstrapped Log-likelihood Ratio Test; VLMRT, Vuong-Lo-Mendell-Rubin Likelihood Ratio Test.

**Table 2. Parameter estimation results of the latent class growth model for exercise self-efficacy.**

| | Parameter | Estimate | SE | t | P |
|---|---|---|---|---|---|
| Class 1 | Intercept | 25.217 | 0.799 | 31.571 | <0.001 |
| | Slope | −0.965 | 0.458 | −2.105 | <0.001 |
| Class 2 | Intercept | 44.561 | 1.070 | 41.652 | <0.001 |
| | Slope | 6.076 | 0.387 | 15.709 | <0.001 |
| Class 3 | Intercept | 30.591 | 0.793 | 38.566 | <0.001 |
| | Slope | 4.193 | 0.336 | 12.491 | <0.001 |

Abbreviation: SE, Standard Error.

**Table 3. Multiple comparisons of the trajectory classes of exercise self-efficacy over time.**

| | (I) Time Point | (J) Time Point | Mean Difference (I − J) | *SE* | *P* | 95% CI of the Difference | |
|---|---|---|---|---|---|---|---|
| | | | | | | LL | UL |
| Class 1 | T1 | T2 | −0.047 | 0.853 | 0.956 | −1.751 | 1.657 |
| | | T3 | 1.391 | 0.883 | 0.120 | −0.374 | 3.156 |
| | | T4 | 3.391 | 0.856 | <0.001 | 1.680 | 5.101 |
| | T2 | T3 | −1.438 | 0.639 | 0.028 | 0.161 | 2.714 |
| | | T4 | 3.438 | 0.652 | <0.001 | 2.135 | 4.740 |
| | T3 | T4 | 2.000 | 0.288 | <0.001 | 1.425 | 2.575 |
| Class 2 | T1 | T2 | −8.540 | 1.108 | <0.001 | −10.739 | −6.341 |
| | | T3 | −13.680 | 1.004 | <0.001 | −15.671 | −11.689 |
| | | T4 | −14.130 | 1.069 | <0.001 | −16.252 | −12.008 |
| | T2 | T3 | −5.140 | 0.526 | <0.001 | −6.183 | −4.097 |
| | | T4 | −5.590 | 0.734 | <0.001 | −7.046 | −4.134 |
| | T3 | T4 | −0.450 | 0.437 | 0.305 | −1.317 | 0.417 |
| Class 3 | T1 | T2 | −3.534 | 0.816 | <0.001 | −5.148 | −1.919 |
| | | T3 | −7.977 | 0.753 | <0.001 | −9.466 | −6.489 |
| | | T4 | −13.451 | 0.770 | <0.001 | −14.975 | −11.928 |
| | T2 | T3 | −4.444 | 0.427 | <0.001 | −5.288 | −3.599 |
| | | T4 | −9.917 | 0.536 | <0.001 | −10.978 | −8.857 |
| | T3 | T4 | −5.474 | 0.338 | <0.001 | −6.142 | −4.806 |

Abbreviation: CI, Confidence Interval; LL, Lower Limit; SE, Standard Error; UL, Upper Limit.

between T3–T4. Consequently, Class 2 was named "High-Efficacy Ascending-Stable Group"; Class 3 started at a moderate level (intercept = 30.591, *P* < 0.001) with continuous improvement (slope = 4.193, *P* < 0.001), and significant differences across all time points. Thus, Class 3 was designated as the "Moderate-Efficacy Continuous Increase Group". The three trajectory subgroups of exercise self-efficacy in patients with multivessel coronary artery disease are shown in Fig 2.

### 3.3. Predictors associated with exercise self-efficacy trajectories

Table 4 provided the results of univariate analysis, in which significant factors associated with different exercise self-efficacy trajectories were gender ($x^2$ = 7.270, *P* = 0.026), educational level ($x^2$ = 21.316, *P* < 0.001), residential location ($x^2$ = 15.367, *P* < 0.001), average monthly household income ($x^2$ = 24.674, *P* < 0.001), exercise habit ($x^2$ = 59.900, *P* < 0.001), diabetes ($x^2$ = 11.033, *P* = 0.004), social support (*F* = 118.753, *P* < 0.001), and anxiety (*F* = 17.534, *P* < 0.001).

Table 5 provided the adjusted values of the odds ratio (OR) and 95% CI derived from the multinomial logistic regression analysis. These results identified significant determinants of distinct exercise self-efficacy trajectories when compared to the moderate-efficacy continuous increase group. The factors influencing the low-efficacy decline group included: having diabetes (*OR*, 3.082; *95% CI*: 1.243–7.641), exercise habits (*OR*, 0.257; *95% CI*: 0.085–0.780), social support (*OR*, 0.514; *95% CI*: 0.418–0.633), and anxiety (*OR*, 1.116; *95% CI:* 1.013–1.230). The factors influencing the high-efficacy ascending-stable group included: high average monthly household income (*OR*, 4.183; *95% CI*: 1.150–15.216), exercise habits (*OR*, 1.711; *95% CI*: 1.206–4.747), and social support (*OR*, 1.333; *95% CI*: 1.199–1.482).

## 4. Discussion

This study is the first to investigate the development trajectory of exercise self-efficacy in patients with multivessel coronary artery disease, and analyzing the influencing factors of different trajectory subgroups. Our findings indicate that there

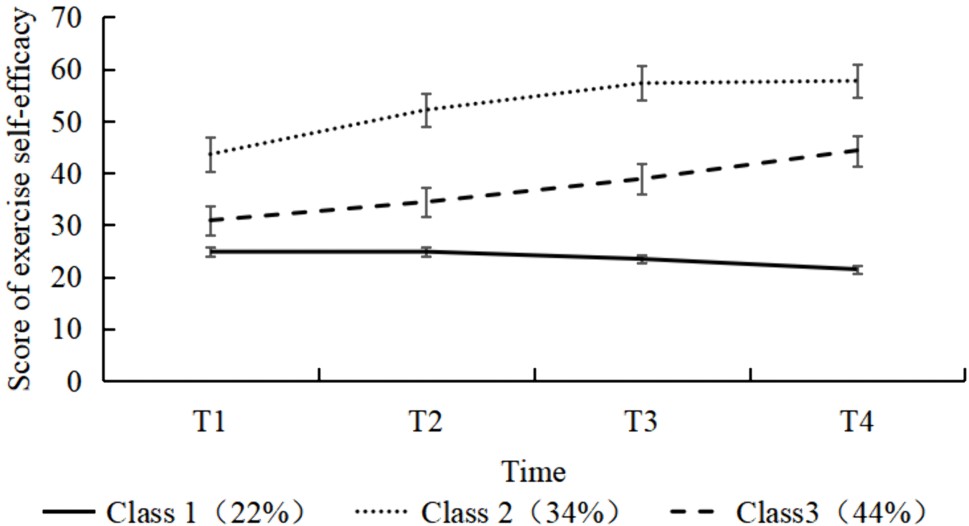

**Fig 2. Three latent classes of exercise self-efficacy trajectories in patients with multivessel coronary artery disease.** This figure illustrates the development of exercise self-efficacy over four time points (T1 to T4) across three latent classes: Class 1 (22%, Low-Efficacy Decline Group), Class 2 (34%, High-Efficacy Ascending-Stable Group), and Class 3 (44%, Moderate-Efficacy Continuous Increase Group). Error bars represent standard errors.

is significant heterogeneity in exercise self-efficacy among patients with multivessel coronary artery disease, revealing three distinct trajectory subgroups: the low-efficacy decline group, the high-efficacy ascending-stable group, and the moderate-efficacy continuous increase group. The factors influencing the exercise self-efficacy trajectory in these patients include average monthly household income, exercise habits, a history of diabetes, social support, and anxiety. While the follow-up points in this study provide insights into the short- to medium-term changes in exercise self-efficacy, the recommendation for a longer follow-up is pertinent. A longer follow-up would facilitate a more thorough understanding of the long-term trajectory of exercise self-efficacy, uncovering potential sustained changes or evolving patterns over an extended duration. Future research with longer follow-up periods is needed to further explore the long-term dynamics of exercise self-efficacy in patients with multivessel coronary artery disease.

The existence of subgroups with distinct exercise self-efficacy trajectories reflects individual differences among patients. Patients with multivessel coronary artery disease in the low self-efficacy decline group (characterized by low baseline self-efficacy and continuous decline over time; key influencing factors: history of diabetes, high level of anxiety, and low income) exhibit low baseline exercise self-efficacy, reflecting limited confidence and ability to engage in exercise rehabilitation at disease presentation. Over time, their self-efficacy declines, potentially because of disease complexity and severity causing frequent rehabilitation setbacks, which in turn leads to a loss of confidence in exercise. Chen et al. found that if patients experienced discomfort symptoms such as dyspnea and chest pain during exercise, they would fear and escape from exercise [30]. Healthcare professionals should prioritize these individuals, engage with them through structured interviews, and gain a comprehensive understanding of their difficulties and concerns during exercise. Simultaneously, rehabilitation programs should commence with low-intensity, simple exercises. For patients with diabetes, a blood glucose adaptation exercise program should be developed, including a ten-minute walk one hour after a meal, and patients should be reminded to monitor their blood glucose before and after exercise to avoid the risk of hypoglycemia or blood glucose fluctuations. For patients with anxiety, mindful breathing was integrated into the exercise process, and two minutes of deep breathing training was performed for every ten minutes of exercise to help relieve the tension during exercise. For low-income patients, the community was provided with exercise material support, free pedometers, home resistance bands and other basic tools, and a simple paper exercise record form was designed. Family members assisted

**Table 4. Comparison of baseline characteristics among different self-efficacy trajectories (*n* = 297).**

| Variables | Group 1 (*n* = 64) | Group 2 (*n* = 100) | Group 3 (*n* = 133) | $\chi^2$/F | P |
|---|---|---|---|---|---|
| Gender, *n* (%) | | | | 7.270 | 0.026 |
| Male | 34(53.1) | 73(73.0) | 91(68.4) | | |
| Female | 30(46.9) | 27(27.0) | 42(31.6) | | |
| Age (years), *n* (%) | | | | 4.458 | 0.108 |
| < 60 | 16(25.0) | 41(41.0) | 45(33.8) | | |
| ≥60 | 48(75.0) | 59(59.0) | 88(66.2) | | |
| Educational level (years), *n* (%) | | | | 21.316 | <0.001 |
| Elementary (≤6) | 27(42.2) | 15(15.0) | 38(28.6) | | |
| Secondary (7–9) | 31(48.4) | 50(50.0) | 61(45.9) | | |
| Higher (>9) | 6(9.4) | 35(35.0) | 34(25.5) | | |
| Marital status, *n* (%) | | | | 4.482 | 0.106 |
| Single | 3(4.7) | 2(2.0) | 11(8.3) | | |
| Married | 61(95.3) | 98(98.0) | 122(91.7) | | |
| Residential type, *n* (%) | | | | 2.888 | 0.236 |
| Living alone | 3(4.7) | 2(2.0) | 9(6.8) | | |
| Living with others | 61(95.3) | 98(98.0) | 124(93.2) | | |
| Employment status, *n* (%) | | | | 2.394 | 0.302 |
| Employed | 38(59.4) | 47(47.0) | 69(51.9) | | |
| Unemployed | 26(40.6) | 53(53.0) | 64(48.1) | | |
| Residential location, *n* (%) | | | | 15.367 | <0.001 |
| Rural | 37(57.8) | 28(28.0) | 61(45.9) | | |
| Urban | 27(42.2) | 72(72.0) | 72(54.1) | | |
| Average monthly household income (RMB), *n* (%) | | | | 24.674 | <0.001 |
| <¥3000 | 32(50.0) | 19(19.0) | 53(39.8) | | |
| ¥3000~5000 | 23(35.9) | 39(39.0) | 48(36.1) | | |
| >¥5000 | 9(14.1) | 42(42.0) | 32(24.1) | | |
| Exercise habit, *n* (%) | | | | 59.900 | <0.001 |
| Yes | 8(12.5) | 73(73.0) | 54(40.6) | | |
| No | 56(87.5) | 27(27.0) | 79(59.4) | | |
| Treatment method, *n* (%) | | | | 1.115 | 0.573 |
| Medication alone | 16(25.0) | 19(19.0) | 32(24.1) | | |
| PCI or CABG | 48(75.0) | 81(81.0) | 101(75.9) | | |
| Family history of coronary heart disease, *n* (%) | | | | 4.487 | 0.106 |
| Yes | 32(50.0) | 56(56.0) | 56(42.1) | | |
| No | 32(50.0) | 44(44.0) | 77(57.9) | | |
| Hypertension, *n* (%) | | | | 4.209 | 0.122 |
| Yes | 47(73.4) | 61(61.0) | 78(58.6) | | |
| No | 17(26.6) | 39(39.0) | 55(41.4) | | |
| Diabetes, *n* (%) | | | | 11.033 | 0.004 |
| Yes | 30(46.9) | 37(37.0) | 32(24.1) | | |
| No | 34(53.1) | 63(63.0) | 101(75.9) | | |
| Hyperlipidemia, *n* (%) | | | | 1.796 | 0.407 |
| Yes | 6(9.4) | 5(6.0) | 6(4.5) | | |
| No | 58(90.6) | 94(94.0) | 127(95.5) | | |

*(Continued)*

**Table 4.** (Continued)

| Variables | Group 1 (*n*=64) | Group 2 (*n*=100) | Group 3 (*n*=133) | ²/F | P |
|---|---|---|---|---|---|
| Stroke, *n* (%) | | | | 2.968 | 0.227 |
| Yes | 6(9.4) | 4(4.0) | 13(9.8) | | |
| No | 58(90.6) | 96(96.0) | 120(90.2) | | |
| Social support score, mean±SD from questionnaire) | 10.11±2.23 | 17.56±3.57 | 14.30±2.91 | 118.753 | <0.001 |
| Anxiety score, mean±SD questionnaire) | 12.82±4.36 | 8.81±4.61 | 9.54±4.25 | 17.534 | <0.001 |

Note: CABG, Coronary Artery Bypass Grafting; PCI, Percutaneous Coronary Intervention.

Group 1 = Low-Efficacy Decline Group; Group 2 = High-Efficacy Ascending-Stable Group; Group 3 = Moderate-Efficacy Continuous Increase Group.

**Table 5. Multinomial logistics regression analysis of exercise self-efficacy.**

| Variables | Group 1 | | | Group 2 | | |
|---|---|---|---|---|---|---|
| | OR | 95% CI | P | OR | 95% CI | P |
| Gender | | | | | | |
| Male | 0.659 | 0.258~1.684 | 0.383 | 0.856 | 0.418~1.753 | 0.671 |
| Female | 1.000 | Referent | | 1.000 | Referent | |
| Educational level (years) | | | | | | |
| Elementary (≤6) | 2.131 | 0.450~10.089 | 0.340 | 1.930 | 0.640~5.822 | 0.243 |
| Secondary (7–9) | 2.699 | 0.647~11.256 | 0.173 | 1.985 | 0.872~4.520 | 0.102 |
| Higher (>9) | 1.000 | Referent | | 1.000 | Referent | |
| Residential location | | | | | | |
| Rural | 1.324 | 0.310~5.653 | 0.705 | 1.711 | 0.632~4.628 | 0.290 |
| Urban | 1.000 | Referent | | 1.000 | Referent | |
| Average monthly household income (RMB) | | | | | | |
| >¥5000 | 1.551 | 0.236~10.210 | 0.648 | 4.183 | 1.150~15.216 | 0.030 |
| ¥3000~5000 | 1.443 | 0.344~6.052 | 0.616 | 2.176 | 0.786~6.025 | 0.135 |
| <¥3000 | 1.000 | Referent | | 1.000 | Referent | |
| Diabetes | | | | | | |
| Yes | 3.082 | 1.243~7.641 | 0.015 | 1.385 | 0.698~2.750 | 0.351 |
| No | 1.000 | Referent | | 1.000 | Referent | |
| Exercise habit | | | | | | |
| Yes | 0.257 | 0.085~0.780 | 0.016 | 1.711 | 1.206~4.747 | 0.013 |
| No | 1.000 | Referent | | 1.000 | Referent | |
| Social supports | 0.514 | 0.418~0.633 | <0.001 | 1.333 | 1.199~1.482 | <0.001 |
| Anxiety | 1.116 | 1.013~1.230 | 0.026 | 0.989 | 0.921~1.061 | 0.752 |

Abbreviations: 95% CI, 95% confidence interval; OR, odds ratio.

Group 1 = Low-Efficacy Decline Group; Group 2 = High-Efficacy Ascending-Stable Group.

Moderate-Efficacy Continuous Increase Group (Group 3) was the referent group.

in recording the daily exercise time or walking distance of patients, and community medical professionals checked and encouraged exercise data through on-site follow-up or telephone communication every week, enabling patients to rebuild their exercise self-efficacy as they progressively achieve success. Patients in the high-efficacy ascending-stable group (characterized by high baseline self-efficacy, rising first and then stable; key influencing factors: high income, good exercise habits, and high social support) demonstrate relatively strong exercise self-efficacy in the early disease stage. As the

time since discharge increases, their self-efficacy initially rises and then stabilizes. The ascending phase likely stems from positive feedback from exercise efforts during rehabilitation, while the stable phase reflects their discovery of a suitable exercise rhythm and mode after achieving a certain fitness level. Bandura notes that improvements in exercise self-efficacy are linked to the successful experiences patients accumulate during exercise, and this positive feedback mechanism aids in continuously boosting their confidence in exercise [10]. For these patients, healthcare professionals can provide advanced exercise recommendations and design personalized advanced exercise plans for high-income patients, such as step-by-step goals from private lessons in the gym to outdoor marathon training. A certification system for sports skills will be established for those who have the habit of exercising, and those who complete events such as swimming or cycling will receive certification badges of rehabilitation institutions. A social support exercise community was built, and an exercise supervision group was formed by family members or friends of patients to carry out collective exercise once a week to help them further improve their physical fitness and encourage them to share exercise experience, thereby inspiring other patients to develop a positive rehabilitation mindset. Compared to the other two groups, the initial exercise self-efficacy level of patients in the moderate-efficacy continuous increase group (characterized by moderate baseline self-efficacy, continued to rise; key influencing factors: stable social support and gradually developing exercise habits) was at a moderate level. Over time, the exercise self-efficacy demonstrated a consistent upward trend, likely due to systematic and gradual rehabilitation guidance. These patients can adhere effectively to the rehabilitation plan and gradually enhance their understanding of their exercise capabilities through ongoing practice. For these patients, healthcare professionals should maintain the existing rehabilitation guidance, develop a monthly assessment system of exercise efficacy based on the dual path of dynamic adjustment and skill development, and automatically adjust the exercise intensity combined with self-efficacy score and exercise endurance test; The exercise form experience class was set up, and a new exercise, such as Baduanjin and rehabilitation exercise, was introduced every month. The exercise companion partner was formed by community volunteers, and the patients in the same group were paired to supervise each other to clock in. At the same time, the intensity and content of exercise were adjusted in real time according to the progress of rehabilitation to ensure the continuous and steady improvement of self-efficacy, ultimately optimizing rehabilitation outcomes.

Our study found that the exercise self-efficacy of patients with multivessel coronary artery disease exhibited population heterogeneity, revealing three distinct trajectory subgroups that differ from those identified in existing research. Liu et al. conducted a trajectory study on the self-efficacy of pelvic floor muscle training among 126 antenatal and postnatal women, utilizing the latent class growth model to categorize participants into four trajectory groups: a high functioning group (56.3%), a development group (9.7%), a block group (22.3%), and a low functioning group (11.7%) [15]. The high functioning group comprised more than half of the total participants, which may be attributed to the differing characteristics of the study populations involved. Liu et al.'s study focused on pregnant women, primarily young individuals [15], whereas the patients in our study included 65.6% individuals over 60 years old, predominantly elderly. Secondly, the population included in this study consisted of healthy individuals in a specific physiological state [15], whereas our study focused on patients with multivessel coronary artery disease. Compared to the healthy population, the disease factors affecting patients with multivessel coronary artery disease significantly impact their exercise confidence. Thirdly, the type of study design can influence the results. This study examined the trend of self-efficacy before and after the intervention of the "Love House" app [15], while our study analyzed the self-efficacy trends of patients with multivessel coronary artery disease after discharge. Additionally, Young et al. conducted a longitudinal survey on self-efficacy in patients with multiple sclerosis, identifying four distinct trajectories using a group-based trajectory model [14]. There are differences in the trajectory subgroups and proportions identified in our study compared to Young et al.'s research [14]. The reasons for these differences may include variations in the number of participants across studies and cultural differences. The cultural backgrounds of different countries and regions can influence individuals' cognitive patterns and health concepts, potentially affecting the measurement of self-efficacy variables and, consequently, the final model fitting results.

Our study found diabetes was an independent risk factor for exercise self-efficacy trajectories in patients with multivessel coronary artery disease, who were more likely to enter the low-efficacy decline group. When coexisting with diabetes, exercise self-efficacy in these patients tends to diminish, which may stem from the increased complexity of their health status introducing unique exercise-related risks. For patients with diabetics, inadequate glycemic control during exercise amplifies exercise-related fear and anxiety, significantly reducing their self-efficacy. Nesti et al. [31] showed that prolonged hyperglycemia in diabetes leads to multi-organ damage, thereby increasing exercise intolerance and anxiety. During physical activity, patients with diabetics are more prone to symptoms like fatigue and hypoglycemia compared to non-diabetic individuals, further undermining their exercise confidence. Additionally, studies indicate that patients with diabetes face substantial long-term disease management pressure, including frequent blood glucose monitoring, strict dietary control, and regular medication taking, which can trigger anxiety and depression [32]. This negative psychology diminishes their enthusiasm for exercise rehabilitation, hindering self-efficacy improvement. For patients with multivessel coronary artery disease who have a history of diabetes, healthcare professionals should develop multidisciplinary personalized exercise plans tailored to their overall condition and alleviate negative emotions by structured interviews, helping them gradually build exercise confidence. Furthermore, enhancing exercise monitoring with trained staff and equipment to promptly address hypoglycemia or cardiovascular events can improve exercise safety, gradually boost self-efficacy, and ultimately enhance rehabilitation outcomes and quality of life.

Our findings indicate that higher income serves as an independent protective factor for exercise self-efficacy trajectories in patients with multivessel coronary artery disease. Patients with greater financial resources were more likely to advance to the high-efficacy ascending-stable group. The economic burden imposed by illness on low-income individuals can exacerbate the psychological stress experienced by both patients and their families [33]. This psychological strain can undermine patients' confidence and motivation for rehabilitation, resulting in resistance to exercise programs and a reluctance to participate in physical activity, ultimately leading to diminished exercise self-efficacy [34]. Moghaddam et al. found a significant correlation between economic status and self-efficacy regarding chemotherapy-related symptoms [35]. Given that the ability to maintain independence and economic stability is a crucial factor in self-efficacy, it is recommended that healthcare professionals develop cost-effective exercise programs tailored to the specific needs of low-income patients. Additionally, they should offer low-cost rehabilitation recommendations. For instance, activities such as walking, jogging, cycling, Baduanjin, and other exercises can be performed in free venues like parks and open spaces within residential areas to ensure patients achieve a sufficient level of physical activity. Furthermore, it is essential to consider the psychological well-being of patients from low-income families by providing understanding and support, enhancing psychological counseling, and improving exercise self-efficacy.

The results of our study indicate that exercise habits serve as a protective factor for the exercise self-efficacy trajectories in patients with multivessel coronary artery disease. Patients with exercise habits are more likely to progress into the high-efficacy ascending-stable group, a finding that aligns with prior research [11,17]. Exercise habits in our study reflect a sustained and diversified pattern of physical activity. Aerobic exercises, with their focus on cardiovascular endurance, lay the foundation for sustaining physical activity, while low-load resistance training enhances muscular strength and endurance, which are essential for performing various physical tasks with confidence. Such an exercise habit ensures that patients not only improve their cardiovascular function but also develop a sense of mastery over their physical capabilities through diverse exercise experiences, which is a core component of exercise self-efficacy. The World Health Organization recommends that maintaining an exercise habit can significantly enhance the function of the cardiovascular system [36]. Regular exercise promotes the development of coronary collateral circulation, which increases blood supply to the myocardium and improves the heart's tolerance to ischemia and hypoxia. Aerobic exercise further enhances myocardial contractility and stroke volume, reducing cardiac workload during daily activities and exercise [36]. For patients with multivessel coronary artery disease, these physiological adaptations enable better tolerance to exercise intensity variations, providing positive feedback that reinforces confidence in physical capability. Exercise habits also profoundly

impact psychological well-being. Regular physical activity can stimulate the secretion of endorphins in the brain, leading to feelings of pleasure and satisfaction while effectively alleviating negative emotions [37]. It is recommended that healthcare professionals should enhance patient education through multi-channel dissemination of exercise rehabilitation knowledge, including safety protocols, benefits, and technique guidance, while utilizing modern technology such as wearable fitness trackers and tele-rehabilitation apps for real-time exercise monitoring and remote guidance to optimize the accessibility and effectiveness of rehabilitation programs, ultimately helping patients develop healthy exercise habits.

This study demonstrated that patients with multivessel coronary artery disease who received a high level of social support were more likely to progress into the high-efficacy ascending-stable group. When patients are encouraged and supported by family and friends, they can boost their confidence in managing the disease and participating in exercise rehabilitation. The result of Zhu et al. [38] showed that social support had an indirect positive effect on patient activation through self-efficacy. The study by Chair et al. [39] demonstrated that social support positively influences exercise self-efficacy, and strong social relationships can help patients develop greater confidence in their ability to exercise, thereby overcoming movement disorders. Good social support can alleviate psychological stress and enhance patients' confidence in overcoming their illness. According to Bandura's self-efficacy theory, social support enhances exercise self-efficacy [10]. In this study, patients with high social support were more likely to accumulate positive feedback from exercise, such as improved physical function, which aligns with the theory's emphasis on environmental factors reinforcing self-efficacy beliefs. Healthcare professionals still need to pay attention to the evaluation and monitoring of patients' social support systems in relation to their exercise rehabilitation. For patients lacking adequate social support, organizing mutual aid groups can significantly enhance their social support. This approach aims to improve the exercise self-efficacy of patients with multivessel coronary artery disease.

The results of this study indicated that the anxiety levels of patients with multivessel coronary artery disease significantly influenced their exercise self-efficacy development trajectory. Patients exhibiting high levels of anxiety were at a greater risk of transitioning into a low-efficacy decline group. The reason may be that anxiety can interfere with a patient's cognition and behavior. When experiencing high levels of anxiety, patients often focus excessively on the negative consequences of their condition, overreact to minor discomfort during exercise rehabilitation, exaggerate perceived risks associated with physical activity, and develop fear [40]. Consequently, their exercise self-efficacy tends to be low. Additionally, anxiety can hinder patients' ability to concentrate and focus on their exercise routines, which negatively impacts the effectiveness of their workouts and further diminishes their self-efficacy. After experiencing cardiovascular events, patients often struggle with negative emotions such as anxiety and depression. These feelings can lead to a decrease in internal motivation and difficulty in adhering to exercise regimens [41]. This suggests that anxiety negatively impacts patients' enthusiasm for exercise, ultimately reducing their self-efficacy in maintaining an active lifestyle. Patients with high levels of anxiety are more likely to exhibit exercise avoidance behavior, which hinders the development and maintenance of their exercise self-efficacy. It is recommended that healthcare professionals prioritize the anxiety levels of patients with multivessel coronary artery disease. They should promptly identify patients with elevated anxiety and implement psychological interventions, such as cognitive behavioral therapy, mindfulness meditation, and music therapy, to alleviate anxiety severity and enhance patients' exercise self-efficacy.

## 5. Limitation

Although this study represented a step forward in the understanding of exercise self-efficacy trajectory patterns considering multinomial heterogeneity, there still existed several limitations. First, the study subjects were drawn exclusively from three tertiary hospitals in Tangshan, which may limit the regional representativeness of the sample and affect the generalizability of the results. Future studies could expand the sample size to include patients from diverse regions and varying medical conditions to enhance the applicability of the findings. Second, the follow-up period was limited to six months, preventing an exploration of the long-term trajectory of exercise self-efficacy. Future studies should extend the follow-up

period to obtain a more comprehensive understanding of exercise self-efficacy trajectories in patients with multivessel coronary artery disease. Third, the study did not include detailed assessments of clinical severity of multivessel coronary artery disease, such as the Gensini score or the extent of stenosis in each diseased vessel. This may restrict a comprehensive evaluation of disease severity and its potential influence on the results. Future studies could include multiple indicators to assess disease severity more thoroughly, such as the Gensini score and left ventricular ejection fraction, to further validate our findings. Fourth, we did not explore potential interactions among predictors (e.g., diabetes × anxiety, social support × income). Future research with larger sample sizes is warranted to investigate these interactions, as they may reveal nuanced differences in how predictors influence exercise self-efficacy across different subgroups, thereby informing more tailored intervention strategies. Additionally, predictors may change over time; however, we only measured predictors in patients with multivessel coronary artery disease at baseline. Future studies should explore the relationship between exercise self-efficacy trajectories and predictors that evolve concurrently over time.

## 6. Conclusions

In conclusion, our study identified three developmental trajectories of exercise self-efficacy in patients with multivessel coronary artery disease. The factors influencing these trajectories included average monthly household income, a history of diabetes, exercise habits, social support, and anxiety. Healthcare professionals should prioritize patients with low income and a history of diabetes. Additionally, they should assist patients with multivessel coronary artery disease in developing exercise habits, reducing anxiety, and enhancing social support to improve exercise self-efficacy and promote early rehabilitation.

## Supporting information

**S1 File. The STROBE checklist is presented in S1 File.**
(DOCX)

**S2 Table. Analysis of systematic differences between included and excluded patients with multivessel coronary artery disease.**
(DOCX)

**S3 Table. Results of 10-fold cross-validation for sensitivity analysis.**
(DOCX)

**S4 Data. The data used in this study are shown in S4 Dada.**
(XLSX)

## Author contributions

**Conceptualization:** Binbin Sun, Jin Wang, Haijiao Xiao, Yutong Wang, Jianhui Wang.

**Data curation:** Binbin Sun.

**Investigation:** Binbin Sun, Haijiao Xiao, Yutong Wang.

**Methodology:** Binbin Sun, Jin Wang, Haijiao Xiao, Yutong Wang.

**Software:** Binbin Sun, Jin Wang, Haijiao Xiao, Yutong Wang.

**Supervision:** Jin Wang, Jianhui Wang.

**Validation:** Jin Wang.

**Writing – original draft:** Binbin Sun, Jin Wang, Haijiao Xiao, Jianhui Wang.

**Writing – review & editing:** Jianhui Wang.

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
