## [Decision Letter · Decision Letter 0]

17 Sep 2025

PONE-D-25-35625Heterogeneous trajectories of exercise self-efficacy and its predictors in patients with multivessel coronary artery disease: a longitudinal studyPLOS ONE

Dear Dr. Wang,

Thank you for submitting your manuscript to PLOS ONE. After careful consideration, we feel that it has merit but does not fully meet PLOS ONE’s publication criteria as it currently stands. Therefore, we invite you to submit a revised version of the manuscript that addresses the points raised during the review process.

We look forward to receiving your revised manuscript.

Kind regards,

Hossein Ali Adineh, Ph.D

Academic Editor

PLOS ONE

Journal Requirements:

“This study was supported by the Medical Science Research Project of Hebei (No.20241512). “

5. Please include a caption for figure 1.

Reviewers' comments:

Reviewer's Responses to Questions

**Comments to the Author**

1. Is the manuscript technically sound, and do the data support the conclusions?

Reviewer #1: Yes

Reviewer #2: Yes

2. Has the statistical analysis been performed appropriately and rigorously? 

Reviewer #1: Yes

Reviewer #2: Yes

3. Have the authors made all data underlying the findings in their manuscript fully available?

Reviewer #1: No

Reviewer #2: Yes

4. Is the manuscript presented in an intelligible fashion and written in standard English?

Reviewer #1: Yes

Reviewer #2: Yes

5. Review Comments to the Author

Reviewer #1: This manuscript identifies three distinct trajectories of exercise self-efficacy among patients with multivessel coronary artery disease using a latent class growth model: low-efficacy, moderate-efficacy, and high-efficacy. The study advances understanding of factors contributing to exercise efficacy by revealing key predictors of these trajectories, such as diabetes, exercise habits, social support, income, and anxiety, thereby emphasizing the need for personalized rehabilitation strategies.

Major concerns:

1. The study population was patients with multivessel coronary artery disease. However, within this population, it is possible that there is a wide range of clinical severity of disease that could confound findings. Did authors consider how clinical severity of coronary artery disease may affect results?

Minor concerns:

1. Abstract – consider briefly defining developmental trajectory of exercise self-efficacy as many readers of PLOS one will be unfamiliar with this terminology.

2. Line 33-34. The line starting with “despite evidence that…” is an incomplete sentence and I believe should be combined with the following one or revised.

3. Line 38: How is optimal level defined?

4. Line 85-89: Not sure how relevant your sample size calculations are, consider removing

5. Line 94. Please define exercise habit. This characteristic ends up being an important factor for exercise self-efficacy so a clear understanding of how this is defined both in the methods and in context of your discussion is crucial.

6. Line 95: Were any other comorbidities in addition to hypertension and diabetes considered? Why only include these?

7. Line 101: Consider providing a brief description of the three dimensions (task efficacy, coping efficacy, and scheduling efficacy) to allow readers to better understand the main measurement outcome of your study.

8. Line 117: What was the training and qualifications required for researchers?

9. Line 119: By investigation here, do you mean specifically survey administration?

10. Line 129: Was the previous longitudinal study looking specifically and exercise efficacy?

11. Table 4 &5: For average monthly income, is the middle category 3000-5000?

12. Table 4: For the social support and anxiety rows, please clarify that these are the mean scores from the questionnaires.

Reviewer #2: • Measurement timing: Do the selected follow-up points (day 3, 1, 3, 6 months) adequately capture changes in exercise self-efficacy, or would a longer follow-up (e.g., 12 months) provide a clearer trajectory?

• Generalizability: Recruitment from three tertiary hospitals in Tangshan may limit external validity. How might regional and healthcare system factors affect the findings?

• Attrition and missing data: How were exclusions and loss to follow-up handled? Were there systematic differences between included and excluded patients that could bias results?

• Trajectory model choice: Both 3-class and 4-class models showed fit. Why was the 3-class solution preferred? A sensitivity check comparing solutions could add confidence.

• Predictor analysis: Were potential interactions (e.g., diabetes × anxiety, social support × income) tested? This may provide deeper insight into subgroup heterogeneity.

• Implications for practice: The intervention suggestions are broad. Can the authors propose more specific, tailored strategies for each trajectory group?

6. PLOS authors have the option to publish the peer review history of their article (what does this mean? ). If published, this will include your full peer review and any attached files.

**Do you want your identity to be public for this peer review?** For information about this choice, including consent withdrawal, please see our Privacy Policy .

Reviewer #1: No

Reviewer #2: No

---

## [Author Response · Author response to Decision Letter 1]

27 Oct 2025

October 23, 2025

Hossein Ali Adineh, Ph.D

Academic Editor

PLOS ONE

Dear Dr. Hossein Ali Adineh,

I am writing on behalf of my co-authors to submit our revised manuscript PONE-D-25-35625, entitled “Heterogeneous trajectories of exercise self-efficacy and its predictors in patients with multivessel coronary artery disease: a longitudinal study” to be considered for publication in PLOS ONE.

We wish to thank you and the two reviewers for your thoughtful comments and suggestions. In this revised version, we have made revisions according to your requirements and the comments from two reviewers. Point-by-point responses to the editor and two reviewers are listed below this letter. To facilitate this discussion, we retyped your comments in italic font and then present our responses.

All authors have read, and approved submission of the manuscript and this manuscript has not been published and is not being considered for publication elsewhere in whole or part in any language. All authors meet the 4 ICMJE criteria for authorship.

Thank you for reviewing and providing your suggestions on improving our manuscript. I am available to address any questions that might arise during the resubmission process. We appreciate the time and effort that you and the editorial reviewers have devoted to considering our manuscript.

Sincerely,

Jianhui Wang, PhD, RN

Nurse Administration Department, Tangshan Gongren Hospital, Tangshan, China.

E-mail: anita30@163.com

PONE-D-25-35625

Authors’ responses to the editor

Comment 1:

Response 1:

We sincerely thank the editor for your careful review. As suggested, we have revised the format of the full text and the file naming to meet the style requirements of PLOS ONE. Thank you again for your helpful suggestions.

Comment 2:

We note that the grant information you provided in the ‘Funding Information’ and ‘Financial Disclosure’ sections do not match.

Response 2:

We apologize for this oversight. In our resubmitted cover letter, we provided the correct funding information, which is “Funding Information

This study was supported by the Medical Science Research Project of Hebei (No.20241512).”

Thank you again for your careful review.

Comment 3:

We note that your Data Availability Statement is currently as follows: [All relevant data are within the manuscript and its Supporting Information files.]

Response 3:

Thank you very much for your detailed review and valuable comments. All raw data underlying figures and analyses are provided in Supporting Information (S4 Data).

Comment 4:

Thank you for stating the following in the Acknowledgments Section of your manuscript:

“This study was supported by the Medical Science Research Project of Hebei (No.20241512).”

Response 4:

Thank you for your careful review. We have removed the descriptions of the funding information in the Acknowledgements Section and manuscripts. According to your suggestion, we have attached the revised statement to the cover letter. Thank you once again for your helpful comments.

Comment 5:

Please include a caption for figure 1.

Response 5:

Page 10, line 227-231:

Thank you for your detailed review. Based on your suggestion, we have added a caption for Figure 1, which is “Fig 1. Three latent classes of exercise self-efficacy trajectories in patients with multivessel coronary artery disease. This figure illustrates the development of exercise self-efficacy over four time points (T1 to T4) across three latent classes: Class 1 (22%, Low-Efficacy Decline Group), Class 2 (34%, High-Efficacy Ascending-Stable Group), and Class 3 (44%, Moderate-Efficacy Continuous Increase Group). Error bars represent standard errors.” And we have modified the format and naming of the figure according to your requirements. Thank you once again for your careful consideration.

Comment 6:

Response 6:

Thank you for your valuable comment. We have carefully reviewed all reviewer comments and checked for any recommendations to cite specific previously published works. Since there were no such specific citations suggested by the reviewers, we have confirmed that all included references are relevant to our study’s content, methodology, and discussions, and we have not added any citations that are not necessary or indicated by the editor.

PONE-D-25-35625

Authors’ responses to the reviewers

(Note: Due to major revision of the manuscript, pages noted by reviewer which refer to the original manuscript differ that those identified in authors responses which refer to the revised manuscript.)

Replies to Reviewer #1:

Major concerns:

Comment 1: The study population was patients with multivessel coronary artery disease. However, within this population, it is possible that there is a wide range of clinical severity of disease that could confound findings. Did authors consider how clinical severity of coronary artery disease may affect results?

Response 1:

We sincerely appreciate the reviewer’s insightful comment regarding the potential confounding effect of clinical severity of disease within the multivessel coronary artery disease population.

We acknowledge that the lack of detailed assessments of clinical severity (e.g., Gensini score, extent of stenosis in each diseased vessel) is a limitation of our current study. To minimize the potential impact of variability in clinical severity as much as possible, we strictly defined the inclusion criteria for “multivessel coronary artery disease” as “≥ 2 major coronary arteries with ≥ 50% luminal stenosis confirmed by coronary angiography” to ensure a relatively homogeneous group in terms of baseline disease extent (Page 4, line 83).

Recognizing this gap, in future research, we plan to incorporate comprehensive measures of clinical severity (e.g., Gensini scoring, detailed angiographic findings) to further investigate the interplay between disease severity and exercise self-efficacy trajectories in this population.

Page 22, line 446 to Page 23, line 450:

According to your suggestion, we have added the following content to the Limitations section of the manuscript, which is “Third, the study did not include detailed assessments of clinical severity of multivessel coronary artery disease, such as the Gensini score or the extent of stenosis in each diseased vessel. This may restrict a comprehensive evaluation of disease severity and its potential influence on the results. Future studies could include multiple indicators to assess disease severity more thoroughly, such as the Gensini score and left ventricular ejection fraction, to further validate our findings.”

Thank you again for helping us strengthen the manuscript.

Minor concerns:

Comment 1:

Abstract – consider briefly defining developmental trajectory of exercise self-efficacy as many readers of PLOS one will be unfamiliar with this terminology.

Response 1:

Page 1, line 4-5:

Thank you for your constructive comments. Based on your suggestions, we have added a brief definition to the development trajectory of exercise self-efficacy to help readers better understand, and to read “The developmental trajectory of exercise self-efficacy refers to the course of change in an individual's belief in their capability to successfully perform exercise-related behaviors over time.” We sincerely thank you for your detailed suggestions from the reader’s point of view. Thank you again for your helpful comments.

Comment 2:

Line 33-34. The line starting with “despite evidence that…” is an incomplete sentence and I believe should be combined with the following one or revised.

Response 2:

Page 2, line 35-37:

Thank you so much for your careful review. According to your comments, we have modified the sentence beginning with “despite evidence that...” to merge with the following sentence, to read “Despite evidence that exercise rehabilitation improves cardiovascular function, exercise capacity, and quality of life [5,6], existing studies indicate that the current state of exercise rehabilitation across various countries is concerning.” Thank you again for your detailed comments.

Comment 3:

Line 38: How is optimal level defined?

Response 3:

Page 2, line 40-41:

We sincerely appreciate the reviewer's valuable comment, which contributes to enhancing the clarity and precision of our manuscript. Regarding the definition of “optimal level” in the original line 38, our intended meaning is that the patients' participation rate is far below what is recommended by cardiac rehabilitation guidelines. To make this clearer, we have revised the relevant content in the manuscript. The original sentence has been modified to “far below the minimum level recommended by cardiac rehabilitation guidelines for effective cardiac recovery” to explicitly indicate that it is the minimum requirement from the guidelines that is not met. Thank you again for your constructed comments.

Comment 4:

Line 85-89: Not sure how relevant your sample size calculations are, consider removing

Response 4:

Page 4, line 92 to Page 5, line 103:

Thank you for your rigorous comments. Regarding the concern about the relevance of the sample size calculations and the suggestion of removing this section, we would like to emphasize that sample size calculation is a crucial component of study design to ensure the statistical validity and reliability of the results. In view of your consideration, we found that the Sample size calculation in the original writing of “2.3. Sample size” did not highlight the correlation with this study. Therefore, in response to the reviewer's doubts about the lack of direct relevance in the study sample size calculation, we have revised the corresponding section in the manuscript to better contextualize the sample size calculation within the overall aims of our study, thereby enhancing the clarity of its importance without removing this key methodological component, which is “The sample size determination was conducted using G*power 3.1 software. Given that our study involved tracking exercise self-efficacy trajectories derived from four repeated measurements, we employed the Single-group repeated-measures analysis of variance algorithm [20]. The nonsphericity correction (ε) was set to 0.5, a value widely utilized in repeated-measures designs when the precise correlation structure between repeated measurements is not fully established in advance. This value accounts for a moderate deviation from sphericity, thus ensuring the analysis remains robust across diverse potential correlation patterns among the four measurement points. The statistical power was set to 0.8, which strikes a balance between the risk of Type II errors and the practicality of recruiting participants, guaranteeing the study a reasonable likelihood of identifying meaningful trajectory patterns if they exist. With 95% confidence intervals (CI), these parameters resulted in a minimum required sample size of 182. Considering a potential 20% attrition rate, a total of 219 participants were deemed necessary to ensure sufficient statistical power for analyzing the exercise self-efficacy trajectories across the four measurement points.” We believe this revision will make the relevance of the sample size calculations more apparent and strengthen the methodological rigor of our study. Thank you again for your careful consideration.

Comment 5:

Line 94. Please define exercise habit. This characteristic ends up being an important factor for exercise self-efficacy so a clear understanding of how this is defined both in the methods and in context of your discussion is crucial.

Response 5:

Page 5, line 114-118:

Thank you for your excellent suggestions. Based on your comments, we have revised the definition of exercise habit, which is “In this study, we defined the participants as having an exercise habit if they engaged in physical activity for a cumulative duration of at least 30 minutes per day on at least three days per week within the three months prior to hospitalization, including aerobic exercises such as walking, jogging, swimming, and cycling, as well as anaerobic exercises mainly consisting of low-load resistance training.”

Page 20, line 383-388:

Additionally, we have also ensured that the discussion of exercise habit in the Discussion section is consistent with this definition, so as to provide a coherent narrative regarding its role in relation to exercise self-efficacy. Based on your comments, we have added: “Exercise habits in our study reflect a sustained and diversified pattern of physical activity. Aerobic exercises, with their focus on cardiovascular endurance, lay the foundation for sustaining physical activity, while low-load resistance training enhances muscular strength and endurance, which are essential for performing various physical tasks with confidence. Such an exercise habit ensures that patients not only improve their cardiovascular function but also develop a sense of mastery over their physical capabilities through diverse exercise experiences, which is a core component of exercise self-efficacy. ” Thank you once again for your careful consideration.

Comment 6:

Line 95: Were any other comorbidities in addition to hypertension and diabetes considered? Why only include these?

Response 6:

Thank you for your detailed comment. To address this, we would like to clarify that we did consider other comorbidities, including hyperlipidemia and stroke, and collected relevant data. We focused on hypertension and diabetes due to their high prevalence in multivessel coronary artery disease patients (hypertension: 63.34%, diabetes: 34.50% in our study) and their well-documented roles as independent risk factors for cardiovascular outcomes and exercise self-efficacy. Hyperlipidemia and stroke were also evaluated but not prioritized for analysis, as their relatively lower prevalence (hyperlipidemia: 6.73%, stroke: 9.16% in our study) limited the statistical power to detect meaningful associations with exercise self-efficacy trajectories (S2 Table and S4 Dat

---

## [Decision Letter · Decision Letter 1]

9 Dec 2025

Heterogeneous trajectories of exercise self-efficacy and its predictors in patients with multivessel coronary artery disease: a longitudinal study

PONE-D-25-35625R1

Dear Dr. Wang,

We’re pleased to inform you that your manuscript has been judged scientifically suitable for publication and will be formally accepted for publication once it meets all outstanding technical requirements.

Kind regards,

Hossein Ali Adineh, Ph.D

Academic Editor

PLOS One

---

## [Editor Report · Acceptance letter]

PONE-D-25-35625R1

PLOS One

Dear Dr. Wang,

I'm pleased to inform you that your manuscript has been deemed suitable for publication in PLOS One. Congratulations! Your manuscript is now being handed over to our production team.

Kind regards,

on behalf of

Dr. Hossein Ali Adineh

Academic Editor

PLOS One